# The Genetics of Spondyloarthritis

**DOI:** 10.3390/jpm10040151

**Published:** 2020-10-02

**Authors:** Roberto Díaz-Peña, Patricia Castro-Santos, Josefina Durán, Catalina Santiago, Alejandro Lucia

**Affiliations:** 1Faculty of Health Sciences, Universidad Autónoma de Chile, Talca 3460000, Chile; 2Inmunología, Centro de Investigaciones Biomédicas (CINBIO), Universidad de Vigo, 36310 Vigo, Spain; patricassan@gmail.com; 3Department of Rheumatology, School of Medicine, Pontificia Universidad Católica de Chile, Santiago 7690000, Chile; jgduran@uc.cl; 4Faculty of Sport Sciences, European University of Madrid, 28670 Madrid, Spain; catalina.santiago@universidadeuropea.es (C.S.); alejandro.lucia@universidadeuropea.es (A.L.); 5Research Institute Hospital 12 de Octubre (‘imas12’), 28041 Madrid, Spain

**Keywords:** spondyloarthritis, ankylosing spondylitis, genetics, genome-wide association studies, polymorphism, ethnic group, HLA-B27, ERAP1, KIR

## Abstract

The term spondyloarthritis (SpA) encompasses a group of chronic inflammatory diseases with common features in terms of clinical presentation and genetic predisposition. SpA is characterized by inflammation of the spine and peripheral joints, and is also be associated with extra-articular inflammatory manifestations such as psoriasis, uveitis, or inflammatory bowel disease (IBD). The etiology of SpA is not completely understood, but it is known to have a strong genetic component dominated by the human leukocyte antigen (HLA)-B27. In the last few years, our understanding of genetic susceptibility to SpA, particularly ankylosing spondylitis (AS), has greatly improved thanks to the findings derived from powered genome-wide association studies (GWAS) based on single nucleotide polymorphism (SNP) arrays. These studies have identified many candidate genes, therefore providing new potential directions in the exploration of disease mechanisms, especially with regard to the key role of the immune system in the pathogenesis of SpA. SpA is a complex disease where genetic variability, environmental factors, and random events interact to trigger pathological pathways. The aim of this review is to summarize current findings on the genetics of SpA, some of which might help to study new treatment approaches.

## 1. Introduction

The term spondyloarthritis (SpA) encompasses a group of chronic inflammatory diseases that exhibit common features in terms of clinical presentation and genetic predisposition. SpA is characterized by inflammation of the spine and peripheral joints, and can also be associated with extra-articular inflammatory manifestations such as psoriasis, uveitis, or inflammatory bowel disease (IBD). The spectrum of SpA includes several conditions, including ankylosing spondylitis (AS), psoriatic arthritis (PsA), reactive arthritis (ReA), IBD-associated (or ‘enteropathic’) arthritis, juvenile spondyloarthritis, and undifferentiated SpA.

Genetic variability, environmental factors, and random events can interact to trigger pathological pathways involved in SpA development [1]. With regard to genetic factors, although the etiology of SpA is not completely understood, it is known to have a strong genetic component, dominated by the human leukocyte antigen (HLA)-B27 gene. The association with HLA-B27 has been known since 1973 [2,3]. In the last few years, our understanding of genetic susceptibility to SpA, particularly AS, has considerably improved due to an increase in the statistical power of genome-wide association studies (GWAS) based on single nucleotide polymorphism (SNP) arrays [4]. The aim of this review is to summarize current state of knowledge on the genetics of SpA. Because AS is the prototype of SpA, many of the sections will focus on this condition. We also highlight potential future research directions.

## 2. Genetic Epidemiology of SpA

It is known that SpA runs strongly within families [5]. The recurrence risk ratio of the sibling or first-degree relative of a patient has been estimated at 80 for AS and 40 for SpA as a whole [6,7]. AS is indeed highly heritable, with studies of disease recurrence conducted in twins showing a heritability for this condition of ~90% [8]. There is also high heritability with regard to the clinical manifestations of the disease [9] (e.g., 40% for radiographic disease severity and 62% for age of symptom onset). Because of the low prevalence of AS, ranging from 0.1% to 1.4% globally [10] (and thus with a very low number of twin pairs that can be assessed in research studies), it could be that the reported high values of heritability are largely due to the small sample size of most studies published in the field. In this regard, recent developments in heritability estimation have been made using SNP data from unrelated cases and controls. Thus, in the UK biobank, the heritability of AS has been estimated at 39.9%, vs. 7.4% for rheumatoid arthritis (RA), 24.1% for Crohn’s disease (CD), 18.6% for ulcerative colitis (UC), and 16.2% for psoriasis. The differences between the heritability estimations reported in twin and control-case studies suggest that there are common SNP variants involved in the pathogenesis of AS that are yet to be identified. The SNPs associated with AS would explain 27.8% of the heritability of this condition, with greatest contribution coming from the major histocompatibility complex (MHC) loci (i.e., 20.4%, vs. 7.4% for non-MHC loci) [11,12].

AS, PsA, and IBD share a common immunopathogenesis. These disorders are collectively associated with HLA class I molecules and have similar clinical features [13]. Although the exact aetiology of PsA remains unclear, cumulative evidence implicates a substantive role for genetic factors [14,15]. PsA is associated with multiple HLA molecules including HLA-B08, HLA-B27, and HLA-B38, while HLA-C06 is specific of psoriasis [16]. Besides MHC, a large number of genetic loci have been described in PsA using GWAS [17,18]. Although the association of IBD-associated arthritis with HLA is weaker compared to the other types of SpA, the role of genetics in this condition is also clear. Concordance in monozygotic twins is 50–75% for CD, although the phenotypic concordance of UC in monozygotic twins is lower (10–20%), suggesting that heritability might be less important [19]. GWAS and their subsequent meta-analyses have improved our understanding of the importance of genetic susceptibility in IBD, with more than 200 loci currently known to be potentially associated with this condition [20]. Identifying predisposing genetic polymorphisms in the context of SpA might provide clues to understanding the pathogenic mechanisms involved in this condition.

## 3. Major Histocompatibility Complex

The genomic map of the human MHC (HLA) spans about 7.6 Mb located on the short arm of chromosome 6 (Figure 1a), and encodes core components of the immune system [21]. The classical HLA loci are labelled -A, -B, and -C (i.e., class I) and -DRB1, -DQB1, and -DPB1 (i.e., class II). HLA class I and II genes encode proteins that play a key role in the immune system by modulating responses to invading pathogens. Thus, both HLA class I and II products are involved in disease resistance and susceptibility. Specifically, the HLA region is known for its association with autoimmune diseases [21]. A hallmark of the MHC is that it contains highly polymorphic genes that encode different antigen-presenting molecules [22], with HLA class I genes showing a higher degree of polymorphism than class 2 genes (Figure 1b). The genes of the HLA class I region encode for a heavy chain of 340 amino acids (Figure 1c), whose extracellular domains (α1, α2, and α3) are encoded by exons 2, 3, and 4, respectively. Of note, the gene encoding β2-microglobulin, the light non-polymorphic chain, is located on chromosome 15.

### 3.1. HLA-B27

AS is one of the best examples of a disease associated with an HLA marker. This association has been demonstrated worldwide [23] and evidence for the specific role in AS of a class I surface antigen encoded by the B locus in the MHC and HLA-B27 comes from both linkage and association studies, and also from research in transgenic animal models [9]. However, identifying the mechanism(s) by which HLA-B27 participates in the pathogenesis of SpA represents a remarkable challenge. Indeed, although it has been almost 50 years since it was first described that the *HLA-B27* gene contributes to SpA susceptibility [2,3], the molecular underpinnings still remain to be clearly elucidated.

The prevalence of HLA-B27 varies between populations and, in general, the prevalence of AS is proportional to the frequency of HLA-B27 [24]. Thus, the prevalence of AS averages ∼8% in Europeans or North Americans, with this condition being rare among African people and Australian Aboriginals. HLA class I molecules present peptides repertoires (derived from the degradation of endogenous proteins) to CD8+ T-cells (also known as ‘cytotoxic T-lymphocytes’) and natural killer (NK) cells. Therefore, peptide binding determines the features of HLA class I molecules and the study of the pathogenic role of HLA-B27 in AS has focused on this phenomenon. More than 200 subtypes of the *HLA-B27* gene have been discovered, some too rare to be investigated for disease associations. The resulting HLA-B27 peptide subtypes differ by amino acid substitutions in the exons 2 and 3 (alpha 1 and alpha 2 domains, respectively) of the peptide-binding cleft, which has six side pockets (conventionally labelled A-F). Differences in antigenic presentation can be largely interpreted in terms of the effect of polymorphisms on pocket interactions.

Not all subtypes are equally distributed in world populations [23]. Thus, AS is relatively more frequently associated with HLA-B*27:05 in nearly all populations of the world, being common in Caucasians and American Indians [25]. Most of the relatively common HLA-B27 alleles (B*27:02, B*27:04, B*27:05, and B*27:07) have been associated with AS [26]: HLA-B*27:02 in Mediterranean populations, HLA-B*27:03 in sub-Saharan/Middle East populations, HLA-B*27:04 in Asian populations, and HLA-B*27:07 in Southeast Asian populations. Two allele subtypes, B*27:06 (common in Southeast Asian populations) and B*27:09 (described only in Sardinia and Southern Italy) appear to be exceptions, as they do not seem to be associated with SpA [27,28].

There are several hypotheses about the influence of the underlying mechanism of HLA-B27 and the involvement of the different B27 alleles on SpA susceptibility [26,29]. The oldest or ‘classic’ theory focuses on canonical functions of HLA-B27 in antigenic presentation. It suggests that the disease arises from the capacity of HLA-B27 to present an ‘arthritogenic peptide’ to CD8+ T-cells (Figure 2A), which ultimately leads to chronic inflammation. Due to the differential disease associations, HLA-B*27:06 and B*27:09 support the theory that a “molecular mimicry” between foreign and self-peptides may unleash a cytotoxic immune response (i.e., of CD8+ T-cells), leading to the autoimmune destruction of self-tissues [30]. As discussed later, some support the idea of the implication of an aberrant peptide presentation, with the influence of different types of HLA-B27 alleles. However, absolute binding preferences of HLA-B27 fail to entirely explain disease association [31]. An alternative mechanism might be that HLA-B27 contributes to SpA through its propensity to form HLA-B27 misfolding or heavy chain homodimerization, thereby inducing endoplasmic reticulum stress caused by the inefficient folding [32], or by interaction of B27 homodimers with receptors of innate immunity [33] (Figure 2B,C, respectively). In addition, alterations in intestinal bacterial communities (i.e., ‘dysbiosis’) have been identified in SpA [34,35], placing the microbiome as an emerging area in SpA [36]. HLA-B27-induced immunological (pro-inflammatory) changes in gut mucosa might be also implicated, with an early and sustained expansion of T-helper 17 (Th17, also known as CD4+) cells, a subset of pro-inflammatory T lymphocytes defined by their production of interleukin 17 [IL-17)] [37]. Perturbation of the interleukin (IL)-23/Th17 axis is a fundamental trigger of chronic inflammation, with Ciccia et al. showing that overexpression of IL-23, but not of IL-17, in the terminal ileum (particularly in Paneth cells) is a pivotal feature of subclinical gut inflammation in AS [38].

### 3.2. Other HLA Class I Genes

Discovery of non-B27 HLA associations with SpA is challenging. The huge linkage disequilibrium (LD) within the HLA region of B27 haplotypes makes it difficult to identify independent contributions of specific allelic variants. However, there is evidence suggesting that the presence of other HLA-B alleles (including B14, B38, B39, B40, and B52) might also confer greater susceptibility to AS [29]. Cortes et al. have identified other HLA-B alleles associated with AS [40] in a large study with 9069 cases and 13,578 controls of European descent. In addition to HLA-B27, several other HLA-B alleles increased disease susceptibility (i.e., the ‘risk’ alleles B*13:02, B*40:01, B*40:02, B*47:01, and HLA-B*51:01), whereas B*07:02 and B*57:01 seemed to be ‘protective’ alleles. Multiple associations of non-HLA-B27 alleles have been replicated in cohorts of patients of Asian ancestry [41] and independent associations with variants in the HLA-A and HLA-C loci have also been observed [40,42].

The prevalence of SpA is lower in Blacks and Africans than in other ethnic groups [43,44]. This could be attributable, at least partly, to a lower frequency of HLA-B27 in the former. However, the influence of HLA markers other than B27 (e.g., different HLA-B alleles) has not been investigated thoroughly in these regions. In this regard, we have conducted studies in sub-Saharan populations (i.e., Burkina Faso, Togo, and Zambia) that showed genetic evidence for an implication of HLA-B*1403 in AS [45,46]. The influence of HLA class I allotypes in AS was homogeneous in the sub-Saharan populations we studied, suggesting a common mechanism of predisposition to AS. The pathogenic behavior of B*14:03 and B*27:05 might be related to a common feature but further research is required to clarify this issue. It would be interesting to investigate the HLA-B distribution among patients with SpA in countries where the influence of B27 on the pathogenesis of this condition is less important, such as Latin American countries [47], and also to study more African countries.

### 3.3. Recognition by NK Cell Receptors

Besides interacting with the T-cell receptor (TCR), HLA class I molecules bind to several other immunomodulatory molecules, including members of the killer immunoglobulin-like receptor (KIR, also known as CD158) family. KIR genes present a great diversity in terms of gene content and expression, and also of allelic polymorphism, encoding both activating and inhibitory receptors [48]. The presence of different combinations of KIR genes can generate inhibitory or activation signals to NK and T-cells, and the effector function is considered to result from the balance of these signals. The contribution of each KIR gene to signaling is not clear, but their relevance in SpA has been supported by different genetic studies (Table 1).

KIRs are unique in terms of their diversity and of their capacity to recognize specific HLA class I allotypes. Notably, KIR3DL1 binds to the HLA-B α1 helix around residues 76–80, with specificity for all Bw4 alleles containing isoleucine at heavy chain residue 80 [49]. KIR3DL1 has been shown to recognize HLA-B27 [50], whereas in vitro-refolded B27 dimers were shown to interact with KIR3DL1 and KIR3DL2 [51]. It actually seems that both KIR receptors (KIR3DL1 and KIR3DL2) are able to bind to HLA-B27 through both the classical beta2m/heavy chain (HC) and the beta2m/free HC homodimers (HC-B27), which are independent of the sequence of the bound peptide [51]. KIR3DL2 is present on the membranes of NK and Th17 cells, and several studies have shown that HC-B27 can interact with KIR3DL2 to promote the survival and growth of both NK and Th17 cells [39,52]. This could activate the IL-23/IL-17 axis to launch the inflammatory reaction in SpA patients, but this mechanism of activation, originally derived from the HLA-B27 misfolding, needs to be characterized.

KIRs and HLA class I molecules might have a modulating effect on SpA development, through a genetic imbalance between activating and inhibitory signals. This could be caused by upregulation of activation or by the loss of inhibition or by a combination thereof, with a relevant role played by specific KIR receptors. There are also studies reporting a genetic influence of KIR genes in PsA (Table 1) [71]. A positive association has been found between the activating KIR2DS1 and KIR2DS2 genes, as well as the inhibitory KIR2DL2 gene and PsA susceptibility. HLA-C1 allotypes (Ser77/Asn80) are ligands for the inhibitory receptors KIR2DL2 and KIR2DL3, as well as the activator receptor KIR2DS2, whereas HLA-C2 allotypes (Asn77/Lys80) are ligands for KIR2DL1 and KIR2DS1. Thus, the absence of ligand for inhibitory KIR (HLA-C1 for KIR2DL2) might involve a lack of inhibitory signal, which would increase susceptibility to PsA. However, despite the fact that there are promising results, to date and from a genetic perspective, most hypotheses are somewhat speculative because of the small number of samples taken into account, at least in the studies performed in PsA.

### 3.4. MHC Class II Region

Several studies have investigated genes involved in antigenic presentation by HLA class I molecules, such as latent membrane protein 2 (*LMP2*) and *LMP7*, and transporters associated with antigen processing *(TAP*) [29]. These findings are interesting as they show how aberrant events in antigenic presentation may explain the pathogenesis of AS. However, the results have proven difficult to replicate. In addition, HLA class II molecules might be also involved in AS development [41]: HLA-DRB1*01:03, HLA-DRB1*11, HLA-DRB1*15:01, HLA-DQB1*02:01, HLA-DQB1*06:02. and HLADPB1*03:01. We conducted a study of high-resolution genotyping in the HLA region enrolling B27-positive patients with AS and healthy controls [72]. The results suggested that HLA-DPA1 and HLA-DPB1 alleles contributed to AS susceptibility. These findings have been replicated subsequently [40,73], providing further evidence for the possible involvement of HLA class II (DP) molecules in the development of AS. HLA-DPA1 and HLA-DPB1 forms the HLA-DP heterodimer, which functions as a receptor for processed peptides derived predominantly from membrane and extracellular proteins, and displays them to CD4+ T-cells. The strongest association has been found for an amino acid substitution in the peptide-binding cleft of HLA-DP [40], suggesting the impact on the peptide repertoire presented by HLA-DP. However, functional experiments are needed to confirm these finding

## 4. Genome Wide Association Studies

### 4.1. Current Status of Variant Discovery

As mentioned above, disease-associated SNP markers explain 27.8% of the heritability of AS, with the greatest contribution (20.45) coming from MHC loci and 7.4% from non-MHC loci [11,12]. However, there are likely common SNP variants involved in AS pathogenesis that remain to be identified. In this regard, GWAS are considered to be one of the primary tools for determining genetic links to diseases [74]. Before the possibility of performing GWAS, “candidate gene studies” based on a priori hypothesis were essentially unsuccessful due to our limited understanding of the genetics of complex traits and yielded many false positive results. The emergence and development of high-throughput genotyping platforms, and analysis methods, enabled to move to the next stage, hypothesis-free GWAS.

The first GWAS involving patients with AS identified two loci, located in endoplasmic reticulum aminopeptidase 1 (*ERAP1*) and IL23 receptor (IL23R) genes, associated with this disease [75]. Since then, there has been an exponential increase in the number of genes associated with AS (Figure 3). To date, over 100 non-MHC loci have been associated with the development of this condition [76]. The largest case–control association study in AS included genotyping of ~130,000 SNPs (Illumina Immunochip, which has incomplete genome-wide coverage) in 8726 patients with AS and 34,213 controls [11]. In this study, Ellinghaus et al. were able to identify 113 AS-associated genome-wide significant variants. Many of the genes implicated can be assigned to different categories according to their biological function and their possible role in disease (Figure 4). GWAS in SpA revealed the potential involvement of mechanisms and pathways that were previously unsuspected, particularly with regard to aminopeptidases or IL23/IL17 pathways. Three M1-aminopeptidases are associated with AS: ERAP1, ERAP2, and puromycin-sensitive aminopeptidase (NPEPPS). ERAP1 and ERAP2 cleave the peptides before their binding to HLA-B27 [77], which alters their function and might change the antigenic pool expressed by HLA-B27 molecules [78]. Moreover, genotypes of ERAP1 and ERAP2 have a considerable impact on the levels of functional enzymes in cells and on the expression of different forms of HLA-B27 on the cell surface [79]. Although the mechanism/s involved remain to be determined, ERAP-inhibition represents a potential therapeutic option [80]. There is a high percentage of genetic variants involved in the IL23 pathway that can influence AS susceptibility [75,81,82]. This has generated great interest in relation to drugs that target this pathway and their potential pharmacogenetic value. However, studies carried out to date were devoid of robustness. Only a randomized double-blind proof-of-concept study has shown an association between IL23R and ERAP1 variants, and the response of patients with AS to treatment with an IL17 inhibitor (i.e., secukinumab) [83]. Another cytokine involved in AS is tumor necrosis (TNF)α, a pro-inflammatory molecule that plays a central role in autoimmune disease pathogenesis. As with IL-23 pathway, variation in TNFα pathway genes could influence the response to (and adverse effects of) TNFα-inhibiting treatments, but the design of these studies is challenging given the multitude of environmental factors and inherent or genetic factors that can affect the drug response. The rest of loci associated to AS can be divided into the following categories: transcription factors and intergenic regions. Among transcription factors is RUNX transcription factor 3 (RUNX2) [12,84], which also has been associated with PsA [85]. RUNX3 plays a prominent role in the development and differentiation of CD8+ T-cells [86], but also has important functions in many other cell types, including chondrocytes, Th1 helper cells, dendritic cells, and NK cells [87]. Thus, understanding the pleiotropic effects of RUNX3 could be relevant to identify key aspects in the pathogenesis of AS, potentially revealing new targets for therapy.

Similar to AS, the strongest genetic signal of susceptibility to psoriasis and PsA comes from the MHC region, mainly HLA-B27 and HLA-C06 [16], although there is a need for a better understanding of the relevance of HLA alleles in this condition. Three GWAS have been reported [90,91,92], identifying 13 regions associated with PsA. More recently, Stuart et al. carried out a GWAS on a very large cohort patients with PsA and unaffected controls (*n* = 1430 and 1417, respectively), and combined their original results with those of previously published data (yielding a total of 3061 cases and 13,670 controls) [18]. They detected 10 additional regions associated with PsA at genome-wide significance. To date, the results of GWAS and subsequent meta-analyses, as well as fine-mapping studies, have allowed identification of 50 susceptibility non-HLA genes associated with PsA [93]. Many of the genes implicated can be grouped into immune/inflammation-related genes, involving biological signalling pathways: IL23/Th17, nuclear factor kappa-light-chain-enhancer of activated B cells (NF-κB), janus kinase (JAK)/signal transducers and activators of transcription (STAT), and mitogen-activated protein kinase (MAPK). The most prominent genes are *IL23A,* TNFAIP3-interacting protein 1 (*TNIP1*), *ERAP1*, *ERAP2*, non-receptor tyrosine-protein kinase (*TYK2*), signal transducer and activator of transcription 4 (*STAT4*), interleukin 12 B (*IL12B*), *RUNX3*, toll-like receptor 4 (*TLR4*), interleukin 13 (*IL13*), *IL23R*, TRAF3 interacting protein 2 (*TRAF3IP2*), and protein tyrosine phosphatase, non-receptor type 22 (*PTPN22*)—some of them shared with other diseases, such as AS, IBD, ReA, RA, and systemic lupus erythematosus. There is a strong need for more GWAS to be conducted in patients with PsA.

Genomic diversity among populations can offer new opportunities to identify genetic variants associated with AS. Recently, Bergström et al. reported a high level of genetic variation restricted by geographical regions [94]. They found an excess of previously undocumented common genetic variation private to the Americas, Southern Africa, Central Africa, and Oceania, which are usually underrepresented populations in GWAS, and absent in the rest of geographical regions (Europe, East Asia, the Middle East, or Central and South Asia). Figure 5 shows the proportion of individuals with different ancestries represented in GWAS carried out in AS. In this Figure, we have included a total of 114,306 individuals extracted from the GWAS Diversity Monitor [95]. There is a disproportionate contribution of data from African American, Afro-Caribbean, and Hispanic or Latin American populations compared to the total number of individuals included in GWAS, with Asians having a significant representation. The percent of people of European ancestry used for initial phase is higher than in replication studies (*p* value < 10^−5^), to the detriment of Asians.

### 4.2. Missing Heritability

Although our understanding of genetic susceptibility to disease has greatly improved thanks to GWAS [74], the loci described in these studies tend to have small effect sizes, being able to explain only a modest proportion of the heritability predicted from traditional genetic epidemiology studies [96]. The case of SpA is not an exception, with only 27.8% of AS heritability being explained to date [11]. Some factors might explain this “missing heritability” [97].

First, it has been suggested that rare and ultra-rare, or low-frequency variants could explain the substantial unexplained heritability of many complex diseases. In this regard, the majority of SNPs included in GWAS are common variants, and thus SNP array-based GWAS are unable to detect ultra-rare variants associated with disease. In addition, few reports have been published that identify rare variants related to SpA. Robinson et al. analyzed the role of rare variants using whole-genome genotyping in 5040 patients with AS and 21,133 healthy controls of European descent [98]. Despite the large sample size, they were unable to identify rare coding variants with a large effect. Only one novel association achieving genome-wide significance was noted at CDK5 regulatory subunit associated protein 1 like 1 (*CDKAL1*) gene. By contrast, studies using a family-based design combined with next-generation sequencing technologies might be more appropriate to identify rare variants. In this regard, O’Rielly et al. showed that the presence of rare syntenic deletions in SEC16 homolog A, endoplasmic reticulum export factor (*SEC16A*) and MAM domain containing 4 (*MAMDC4*) genes increased susceptibility to Axial SpA in family members who carried the HLA-B27 allele [99]. In addition, rare variants have been identified in insulin receptor substrate 1 (IRS1) [100], ankyrin repeat and death domain containing 1B (*ANKDD1B*) [101], and triggering receptor expressed on myeloid cells like 2 (*TREML2*) [102]. However, these variants have not been robustly replicated or studied from a functional point of view. Collectively, these reports suggest that rare variants likely contribute to disease pathogenesis, but highlight the need for more studies on additional rare variants. Structural variants, such as copy number variations (CNV), deletions, and inversions, could be important contributors to complex diseases [103], but they have been poorly investigated. In SpA, a genome-wide microarray study performed in a large family with AS revealed segregation of the UDP glucuronosyltransferase family 2 member (*B17UGT2B17*) gene CNV among all affected family members [104]. In another study, Jung et al. detected 227 CNV regions significantly associated with the risk of AS [105]. The identification of association(s) between rare/ultra-rare or low-frequency variants, even structural variants and a given disease phenotype, should become more tractable using the GWAS approach, especially when whole-genome sequencing (WGS) will become cheaper.

The great challenge in detecting complex (i.e., gene by gene and gene by environmental) interactions makes it difficult to fully explain the heritability of complex traits at the moment. Interactions between alleles at different loci, namely epistasis, could be an important component of the genetic architecture of complex traits [106]. In AS, the existence of an interaction between the ERAP1 SNP rs30187, HLA-B27, and HLAB*40:01 alleles has been demonstrated [40,89]. Similar interactions have been described for Behçet’s disease (BD) and HLA-B*51 [107], psoriasis and HLA-C*06:02 [108], and IBD and HLA-C07 [109]. All of these disorders share genetic susceptibility factors. The finding is biologically interesting, since ERAP1 trims amino terminal residues of precursor peptides to an optimal length in the endoplasmic reticulum, for HLA class I loading. HLA-B27 positive and negative AS cases differ in their association with the ERAP1 gene [89], so it is possible that ERAP1 might be involved in the generation of autoimmunogenic peptides prior to HLA-B27 assembly and peptide presentation. In addition to this finding, no gene by gene interaction study at the whole-genome level has been yet published. Like most rheumatic diseases, SpA is a multifactorial condition where many genetic factors and a high number of diverse environmental factors are involved in disease development [8]. Genetic susceptibility to SpA should be analyzed in the context of environmental risk factors, since different subsets of genes could play relevant roles depending on the risk environment. However, these environmental exposures are not easy to measure and, particularly in SpA, they have been poorly identified beyond microbial agents. In addition, because environmental exposures are not constant over time, a follow-up of SpA patients is needed. Moreover, the expression of genetic variants is modified by environmental factors and the significance of ethnicity in genetics is controversial [110].

## 5. Conclusions and Future Directions

GWAS approaches have increased our understanding of the role of genetic factors in the susceptibility to SpA. Yet, a remaining challenge is to identify those variants that are responsible for the unexplained (or ‘missing’) heritability. Besides the influence of gene by gene and gene by environment interactions, as well as copy number/rare variants, some authors have suggested that SNPs account for most of the ‘missing’ heritability in some health or disease traits [111,112]. In this regard, the benefits of GWAS for identifying new SNPs associated with disease phenotypes are undeniable [74] and increasing the sample size of this type of studies should facilitate the identification of new loci associated with SpA. Notably, sample sizes of over 1 million participants are being included for some conditions (e.g., insomnia [113]) and international collaborations are needed to include large patient cohorts. Efforts underway in many populations worldwide [94,114] should lay the foundation to uncover rare variants that are specific to some ethnic groups or populations, and to study their association with complex phenotypes (an example of which is SpA). On the other hand, advances in NGS techniques and in statistical approaches will allow the integration of multiple “-omics” technologies (transcriptomic, epigenomic, proteomic, metabolomic, and microbiome profiling), and thus to improve the current knowledge on SpA [115].

Approximately one third of the genetic risk in AS has been explained and, except for ERAP1, IL23R, or RUNX3, further studies are needed to confirm the associations discovered, and also to define their involvement in the disease process. Similarly, there is a need for comprehensive case-control studies analyzing the association of all KIR genes with AS, together with a complete HLA class I typing and a rigorous clinical characterization of patients—including also their response to treatment. On the other hand, a better understanding of the genetics of SpA will allow identifying genes that encode proteins representing potential therapeutic targets in SpA. In this regard, the GWAS performed up to date have revealed previously unsuspected players like ERAP1 or IL23/IL17 pathways, which have led to the development of drugs targeting aminopeptidases like ERAP1 and IL-23 pathway inhibitors, respectively. Another factor to consider, which has not been addressed in this review, is the functional mechanisms underpinning genetic associations. This is a challenge in itself, particularly in the case of SpA, where the journey is just at its beginning. Of note, the majority of disease-associated loci are located in non-coding regions of the genome, suggesting a regulatory effect of these variants on gene expression.

Beyond gene identification, GWAS data have enabled a wide range of applications, including development of polygenic risk scores (PRS). The effects of individual SNP markers are limited, but collectively they provide meaningful insights into underlying pathways and contribute to models of risk-stratification for some common diseases [116]. It has been proposed that it is time to consider the inclusion of polygenic risk prediction in clinical care [117]. Rostami et al. found that a genetic risk score based on 110 susceptibility SNPs had a slightly higher ability to predict AS risk than HLA-B27 testing alone [118], albeit the improvement associated with the proposed PRS was small and of uncertain clinical value. Recently, Knevel et al. assessed the conversion of genotype information prior to clinical visit into an interpretable probability value for inflammatory arthritis, including SpA [119]. They demonstrated that genetic data might discriminate different diseases associated with similar clinical signs and symptoms. Thus, PRS are of potential use in early diagnosis or prediction of likelihood of the development of SpA, although further research is required. As more powerful GWAS are performed, future PRSs will allow for a more accurate risk stratification, with integration of familial and environmental risk factors, thereby conforming a global risk score that will improve the prediction of individual risk. In a rheumatology setting, genetic information adds value to the clinical information obtained at the initial encounter, even when serologic data are also available [120]. However, it will be necessary to ensure that all ethnic groups have access to genetic risk prediction, which will require undertaking or expanding GWAS in non-European ethnic groups. Otherwise, the clinical use of PRS might actually contribute to increase health disparities [120].

## Figures and Tables

**Figure 1 jpm-10-00151-f001:**
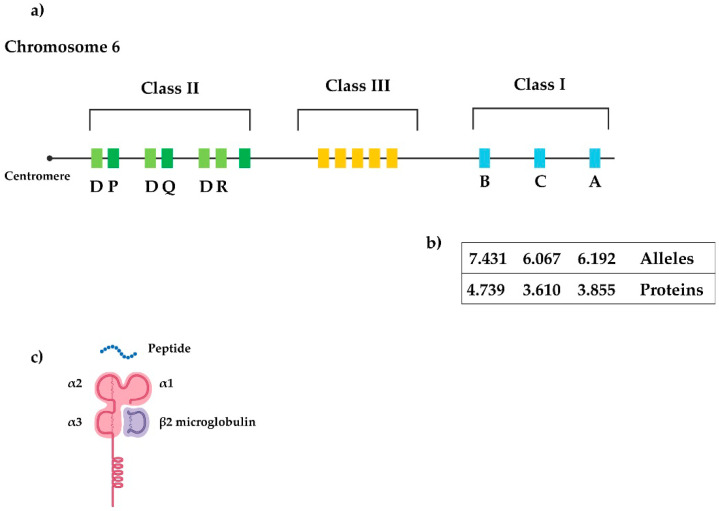
Schematic representation of the human leukocyte antigen (HLA) region (**a**), polymorphism in HLA class I genes (**b**) and structure of an HLA class I molecule (**c**).

**Figure 2 jpm-10-00151-f002:**
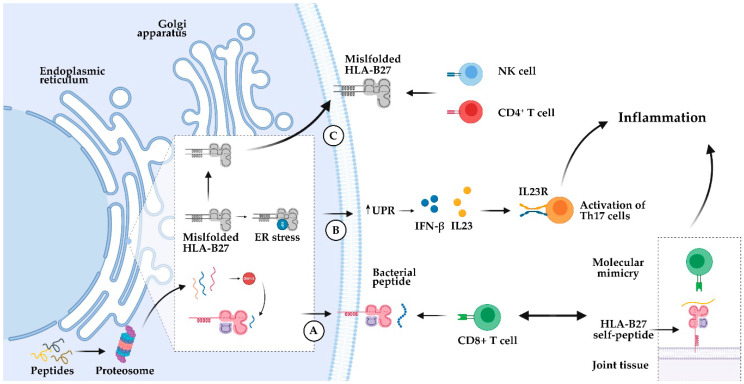
Hypotheses explaining the influence of HLA-B27 in ankylosing spondylitis (AS). According to the ‘classic’ arthritogenic peptide hypothesis, molecular mimicry between foreign and self-peptides could unleash CD8^+^ T-cell cross-reactivity [30], leading to AS (**A**). In turn, the HLA-B27 misfolding hypothesis enunciates that the accumulation of incompletely-assembled HLA-B27 molecules in the endoplasmic reticulum (ER) causes a proinflammatory unfolded protein response (UPR) that leads an increased production of interleukin (IL)-23, together with activation of T-helper (Th17) cells [26] (**B**). HLA-B27 has a propensity to form homodimers that can be recognized by specific receptors expressed on the surface of natural killer (NK) and CD4+ T-cells [33], increasing the expression of the proinflammatory cytokines IL17 and IL23 [39] (**C**).

**Figure 3 jpm-10-00151-f003:**
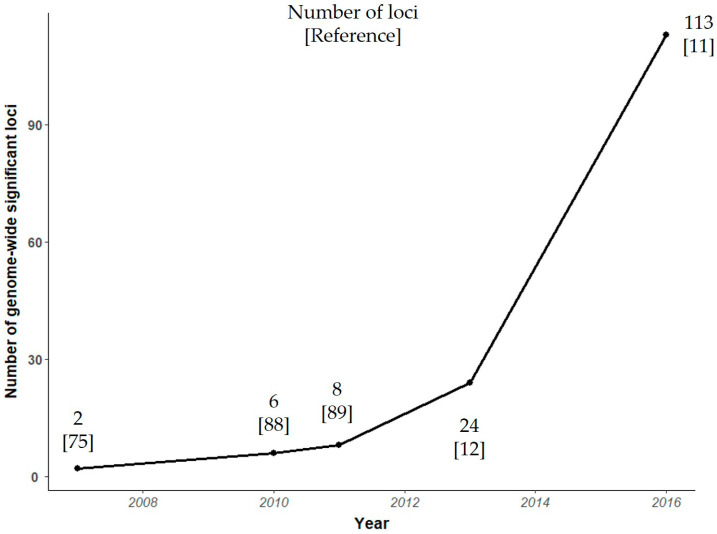
Genome-wide association studies in ankylosing spondylitis (AS) [11,12,75,88,89].

**Figure 4 jpm-10-00151-f004:**
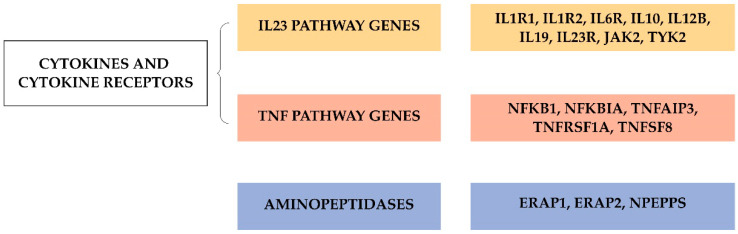
Associated genome-wide association studies (GWAS) genes grouped by categories in ankylosing spondylitis (AS). Abbreviations: ERAP, endoplasmic reticulum aminopeptidase; IL1R1, interleukin-1 receptor 1; IL2R2, interleukin-2 receptor 2; IL6R, interleukin 6 receptor; IL10, interleukin 10; IL12B, interleukin 12B; IL19, interleukin 19; IL23R, interleukin 23 receptor; JAK2, janus kinase 2; NFKB, nuclear factor kappa-light-chain-enhancer of activated B cells 1; NFKBIA, nuclear factor kappa-light-chain-enhancer of activated B cells inhibitor alpha; NPEPPS, puromycin-sensitive amino peptidase; TNFAIP3, TNF alpha induced protein 3; TNFRSF1A, TNF receptor superfamily member 1A; TNFSF8, TNF superfamily member 8; TYK2, non-receptor tyrosine-protein kinase.

**Figure 5 jpm-10-00151-f005:**
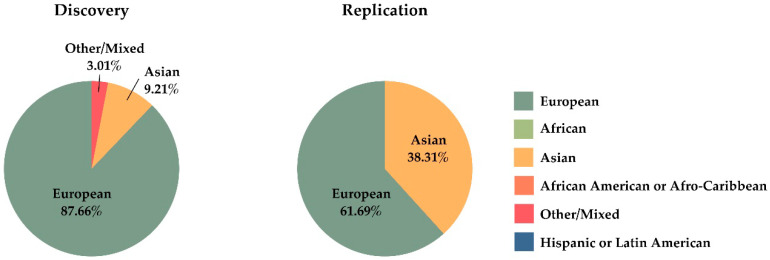
Representation of different ethnic groups in genome-wide association studies carried out in ankylosing spondylitis.

**Table 1 jpm-10-00151-t001:** Basic research studies showing the associations of different types of killer immunoglobulin-like receptor (KIR) with spondyloarthritis (SpA).

Reference	Participants, Main Assessments	Conclusions
[53]	366 patients with PsA and 299 controls. KIR and HLA genotyping.	The activating KIR2DS1 and/or KIR2DS2 genes were more frequent in patients, showing their association with disease risk, but only when HLA ligands for their homologous inhibitory receptors, KIR2DL1 and KIR2DL2/3, were missing.
[54]	220 patients positive for psoriasis vulgaris, 75 also diagnosed as positive for PsA, and 90 controls. KIR genotyping.	The activating KIR2DS1 gene was more frequent in patients with PsA, compared to those with psoriasis negative for PsA, and to unaffected controls.
[55]	396 patients with psoriasis and 372 controls. Psoriasis vulgaris without joint symptoms was diagnosed in 241 patients, guttate (or ‘eruptive’) psoriasis in 80 patients, and PsA in 75 patients. KIR and HLA-C genotyping.	There was a trend towards a higher KIR2DS1 gene frequency among patients with PsA.
[56]	Two HLA-B27–positive Caucasian populations were selected (Spain: 71 patients with AS and 105 controls; and Azores, Portugal: 55 patients with AS and 75 controls). HLA-B and KIR3DS1/3DL1 genotyping	The activating KIR3DS1 gene was associated with AS compared with B27 controls, whereas the inhibitory KIR3DL1 gene was decreased in patients with AS compared with B27 controls. The effect of KIR3DL1 (protection) or KIR3DS1 gene (susceptibility) on AS might stronger when the corresponding ligand Bw4-I80 is present.
[57]	Two HLA-B27–positive Asian populations were selected (China: 42 patients with AS and 30 controls; and Thailand: 30 patients with AS and 16 controls)	KIR3DS1, KIR2DS5, and KIR2DL5 genes were more frequent in patients with AS. The frequency of 3DL1/3DL1 and 3DL1/3DS1 genotypes was lower and higher in patients, respectively.
[58]	200 patients with AS and 405 controls. KIR genotyping.	No differences in KIR genotype frequencies between patients and controls.
[59]	83 patients with AS and 107 controls, all HLA-B27–positive. KIR3DL1 and KIR3DS1 subtyping (genes and alleles).	The frequency of the inhibitory KIR3DL1 gene was lower in patients than in B27 controls. The KIR3DL1 gene was negatively associated with AS at the expense of alleles encoding functional receptors (KIR3DL1*F) but not of KIR3DL1*004 (not functional).
[60]	115 patients with AS and 119 controls, all HLA-B27–positive. KIR and HLA-C genotyping. KIR and HLA-C genotyping.	The frequency of the inhibitory KIR2DL1 and KIR2DL5 gene was higher in patients than in controls.
[61]	270 patients with AS and 435 controls, all HLA-B27–positive. KIR3DL1 and KIR3DS1 subtyping (genes and alleles), and HLA-B genotyping.	The activating KIR3DS1*013 allele was more frequent in patients independent of the presence of the HLA-Bw4I80 epitope, whereas the presence of inhibitory allotypes such as KIR3DL1*004 demonstrated a negative association in patients in the presence of HLA-Bw4I80.
[62]	35 patients with AS and 200 controls. KIR and HLA genotyping.	The telomeric KIR2DL5A, KIR2DS1, and KIR3DS1 genes were more frequent in patients compared to controls. KIR3DL1/Bw4I80 and KIR2DS1/C2 compound genotypes showed association with AS.
[63]	60 patients with AS and 60 controls. KIR genotyping.	The activating KIR3DS1 gene was more frequent in patients than in controls. The frequency of KIR3DL1/KIR3DL1 genotype was lower in patients than in controls.
[64]	110 patients with AS, 86 patients with Behçet disease, and 154 controls. KIR genotyping.	Compared with controls, the frequency of the inhibitory gene KIR3DL1 was lower in patients with AS and also in those affected by uveitis.
[65]	678 patients with PsA and 688 controls. KIR and HLA genotyping.	The activating KIR2DS2 gene was more frequent in patients than in controls.
[66]	176 patients with AS and 435 controls. KIR genotyping.	The frequency of the KIR2DS1 and KIR3DS1 genes was higher in patients than in controls.
[67]	200 patients with PsA and 200 controls. KIR and HLA genotyping.	The inhibitory KIR2DL3 gene was more frequent in patients than in controls, whereas the frequency of the inhibitory KIR2DL5 gene was lower.
[68]	Successful genotyping in 392 patients with PsA, 260 patients with cutaneous psoriasis and with no arthritis, and 371 controls. KIR3DL1 subtyping (alleles).	The non-functional KIR3DL1 allele was associated with psoriatic diseases (i.e., higher frequency in both types of patients compared to controls).
[69]	653 patients with AS and 952 controls. KIR and HLA-C genotyping.	The frequency of the inhibitory KIR2DL5 gene was lower in patients. In addition, KIR2DL5 combined with the HLA-C1/C2 heterozygous genotype showed a protective effect against AS.
[70]	138 patients with ReA and 151 controls. KIR and HLA-C genotyping.	The inhibitory KIR2DL2 and KIR2DL5 genes were less frequent in patients than in controls. The activating KIR2DS1 alone or in combination with the HLA-C1C1 genotype was associated with susceptibility to ReA, whereas KIR2DL2 in combination with the HLA-C1 ligand was associated with protection against ReA.

Abbreviations: HLA, Human Leukocyte Antigen; PsA, psoriatic arthritis; ReA, reactive arthritis.

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
