# Peer review of "The Genetics of Spondyloarthritis"

_jpm, 2020, doi:10.3390/jpm10040151_

Round 1
Reviewer 1 Report
The review article proposed by Diaz-Pena and colleagues is a well-written manuscript focusing on the genetics of spondyloarthritis (SpA), in particular on Ankylosing Spondylitis.
Overall the review covers all the different angles about the role of Genetics in the development of SpA and it is of interest to the SpA field.
However, there are some points to clarify and sections to modify before being considered for publication in order increase the value of the manuscript. Please see below:
GENERAL COMMENT: In Section 1, Introduction, AS is presented as prototype of SpA and 90% of the manuscript covers AS. Agreed with ASAS 2009 SpA classification criteria (Baraliakos Ann Rheum Dis 2009) regarding the spectrum of SpA, I would expect a bit more on the role of Genetics on the other conditions, in specific PsA and IBD. Please include a section focusing on these.
SPECIFIC COMMENTS: In Section 3 paragraph 3.1 entitled HLAB27: In addition to the three canonical theories on AS, I suggest authors to include also the "new" and emerging theory on the intestinal dysbiosis caused by HLA-B27 and the activation of the Th17 cells with the IL-17/IL-23 axis. (REF, Ciccia F et al Arthritis Rheum 2009)
KIR section is extremely detailed, with an enclosed table with all the research studies on KIRs. As showed in REFs 52 and 55, KIRs have been investigated also in PsA. This strength the idea to mention other SpAs as PsA, in the manuscript as stated in my General Comment.
In Section 4, paragraph 4.1, please add a part on PsA GWAS, citing Stuart PE et al AJHG, 2015, GWAS on PsA and psoriasis.
In Section 5 "Conclusions and future directions" the authors should reinforce the concept on the relevance of genetics studies in order to define credible targets and pathways to develop new drugs.
Author Response
- In Section 1, Introduction, AS is presented as prototype of SpA and 90% of the manuscript covers AS. Agreed with ASAS 2009 SpA classification criteria (Baraliakos Ann Rheum Dis 2009) regarding the spectrum of SpA, I would expect a bit more on the role of Genetics on the other conditions, in specific PsA and IBD. Please include a section focusing on these.
R: Thank you for the comment. According to the Reviewer’s suggestion, we have now included more information regarding the role of genetics on the other conditions, i.e., PsA and IBD. Please see revised manuscript, pages 2-3, lines 75-86 and page 10, lines 389-407.
- In Section 3 paragraph 3.1 entitled HLAB27: In addition to the three canonical theories on AS, I suggest authors to include also the "new" and emerging theory on the intestinal dysbiosis caused by HLA-B27 and the activation of the Th17 cells with the IL-17/IL-23 axis. (REF, Ciccia F et al Arthritis Rheum 2009)
R: Done. Please see revised manuscript (Discussion section, page 5 (lines 200-208).
- KIR section is extremely detailed, with an enclosed table with all the research studies on KIRs. As showed in REFs 52 and 55, KIRs have been investigated also in PsA. This strength the idea to mention other SpAs as PsA, in the manuscript as stated in my General Comment.
R: Thank you for the comment. We have now included more information regarding PsA, as mentioned above.
- In Section 4, paragraph 4.1, please add a part on PsA GWAS, citing Stuart PE et al AJHG, 2015, GWAS on PsA and psoriasis.
R: Done (please see revised manuscript, page 10, lines 389-407, and new reference [18]).
- In Section 5 "Conclusions and future directions" the authors should reinforce the concept on the relevance of genetics studies in order to define credible targets and pathways to develop new drugs.
R: Done. Please see revised manuscript, pages 12-13, lines 515-519.
On behalf of all coauthors, many thanks for this insightful review.

Reviewer 2 Report
This is a narrative review about the genetics of spondyloarthritis. The authors reported the current evidences regarding the role of major histocompatibility complex and single nucleotide polymorphisms in the genesis of spondyloarthritis. Although limited by its narrative design, the paper is interesting, well-written and comprehensive. I only suggest the authors to correct some minor mistakes in the paper (eg: Page 8, line 271. Please correct “[…] 27.8 of […]” with “[…] 27.8% of […]”; Page 12, line 482. Please correct “[…] inclusion of polygenic risk prediction in in clinical care […]”).
Author Response
- I only suggest the authors to correct some minor mistakes in the paper (eg: Page 8, line 271. Please correct “[…] 27.8 of […]” with “[…] 27.8% of […]”; Page 12, line 482. Please correct “[…] inclusion of polygenic risk.
R: Thank you for the comment.
We have gone in depth through the entire manuscript for a thorough linguistic and technical edition, and corrected these and other minor (technical/editorial/linguistic) mistakes. Information on polygenic risk is included in the last session.

Round 2
Reviewer 1 Report
The authors addressed my points.
Good job